# Computationally efficient reductions between some statistical models

**Mengqi Lou**                                                    MLOU30@GATECH.EDU
*School of Industrial and Systems Engineering, Georgia Tech*

**Guy Bresler**                                                   GUY@MIT.EDU
*Department of Electrical Engineering and Computer Science, MIT*

**Ashwin Pananjady**                                              ASHWINPM@GATECH.EDU
*Schools of Industrial and Systems Engineering and Electrical and Computer Engineering, Georgia Tech*

**Editors:** Gautam Kamath and Po-Ling Loh

## Abstract

We study the problem of approximately transforming a sample from a source statistical model to a sample from a target statistical model without knowing the parameters of the source model, and construct several computationally efficient such reductions between canonical statistical experiments. In particular, we provide computationally efficient procedures that approximately reduce uniform, Erlang, and Laplace location models to general target families. We illustrate our methodology by establishing nonasymptotic reductions between some canonical high-dimensional problems, spanning mixtures of experts, phase retrieval, and signal denoising. Notably, the reductions are structure-preserving and can accommodate missing data. We also point to a possible application in transforming one differentially private mechanism to another.[1]

**Keywords:** Mixture of experts model, phase retrieval, signal denoising, differential privacy

## Acknowledgments

ML and AP were supported in part by the NSF under grants CCF-2107455 and DMS-2210734 and by research awards/gifts from Adobe, Amazon, and Mathworks. GB gratefully acknowledges support from the NSF Career award CCF-1940205. AP is grateful to Juba Ziani for helpful discussions and pointers to the literature on differential privacy. We are grateful to the Simons Institute program on Computational Complexity of Statistical Inference, during which our discussions were initiated.

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
