# OpenReview forum: "Computationally efficient reductions between some statistical models"
_algorithmiclearningtheory.org/ALT/2025/Conference — ALT 2025_

### Official Review · Reviewer_DPEr · 2024-11-06
**Important problem for statistics and learning theory, clear explanation, proofs are sound**

**Rating:** 9
**Confidence:** 4

**Review:**

This paper studies the problem of transforming a sample (approximately) from a source model to a target model, without knowing parameters of the source model. The authors subsequently establish reductions between various important high-dimensional statistical problems. The arguments and proof techniques are sound.

Clarity:

- The paper is clear and concise. I did not come across a missing definition. Section 1 and 2 are well written.

Originality:

- To the best of my knowledge, the paper studies a broader question than those usually tackled by papers concerning average-case reductions. The contributions are original, and the consequences of their base findings extend results in other papers.

Significance:

- The problem the authors tackle is significant, and dates back to the early days of statistics. I believe the results presented in this paper can/will be used to evidence computational hardness in a variety of new high-dimensional statistical problems (or at least, concerning different base distributions).

Recommendations and Questions:

- In Section 5.1, I recommend citing "Statistical-Computational Tradeoffs in Mixed Sparse Linear Regression" by Arpino, Venkataramanan (COLT 2023), as the authors in that paper study computational hardness in the mixtures of linear regressions (MLR) and phase retrieval (PR) problems as well (although only in the sparse signal regime). From my understanding, they use the trivial (noiseless, infinite SNR) reduction from MLR to PR to evidence computational hardness of sparse phase retrieval through low-degree methods, and hence your reduction which allows for non-infinite SNR seems to help extend that work.
- How do your results relate to the paper: "Hypothesis testing with low-degree polynomials in the Morris class
of exponential families" by Dmitriy Kunisky? From my understanding, they also establish a partial ordering of distributions in a certain model with respect to computational hardness (via the low-degree method). Would be good to cite.

**Paper Award:**

Yes

---

### Official Review · Reviewer_DQo7 · 2024-11-11

**Rating:** 7
**Confidence:** 4

**Review:**

*Summary.* The paper studies computationally efficient approximate reductions between a few distribution families with TV deficiency in the non-asymptotic regime. The goal is to construct a randomized mapping which transforms a sample from a distribution $u_\theta$ with unknown $\theta$ to a sample whose distribution is close to $\nu_\theta$. The computation efficiency is measured in terms of the number of samples generated from a reference distribution and the number of pointwise evaluations under $\nu_\theta$. The main result of the paper is the construction of reductions from distribution families including Laplace, uniform and Erlang location models to arbitrary high-dimensional product distribution under regularity conditions. The paper also describes applications of the reductions in regression and differential privacy (DP).

*Significance* The paper studies reductions between canonical distribution families, which is of interest to the statistics and learning theory community. I don't follow the line of work closely, but the introduced problem of approximate reduction between distribution is novel to the best of my knowledge. The explicit construction of a pure DP mechanism distribution that is close in TV to Gaussian mechanism might be of interest to the DP community.

The paper is nicely written. I didn't examine all the proofs closely but the technical results seem correct based on the discussions and sketches of proof ideas.

*Comments*
The main reason that I didn't give a higher score for the paper is about the general significance of the contribution. The results in the paper are specific to a set of source distributions. Can the authors comment on whether a signed kernel with small error can be constructed for general source distributions?

Could the authors comment on how good the reductions are? For examples, is there a lower bound on the number of oracle calls to the pointwise evaluation on the target model?

While the paper list a few applications, the reduction approach doesn't seem to lead to new results in the described applications.

Corollary 8: It seems the accuracy here is even worse than the sum of the variance of the original Laplace distribution and the targeted Gaussian distribution. So the usefulness of the result is less clear.

==========Post rebuttal=====
Thanks for addressing my questions. My score is increased accordingly.

**Paper Award:**

No

---

### Official Review · Reviewer_e8Br · 2024-11-11

**Rating:** 6
**Confidence:** 2

**Review:**

This submission concerns the construction of transformations between statistical experiments. Suppose we have two experiments about the same underlying parameter:
$(\mathbb{X}, \lbrace \mathcal P_{\theta}\rbrace_{\theta\in\Theta})$ and
$(\mathbb{Y}, \lbrace \mathcal Q_{\theta}\rbrace_{\theta\in\Theta})$.
Here $\mathbb{X}$ and $\mathbb{Y}$ are data spaces and $\mathcal{P}, \mathcal{Q}$ data distributions. The basic problem is to produce a randomized mapping $K$ such that, if $X\sim \mathcal P_{\theta^*}$, then $K(X)$ is approximately distributed according to $\mathcal{Q}_{\theta^*}$. Crucially, the mapping does not access $\theta^*$. This is a classical setup in statistics.

The submission gives a general transformation based on rejection sampling. The algorithm assumes access to some function $S(y\mid x)$, a signed kernel between $\mathbb{X}$ and $\mathbb{Y}$. They show that, when this kernel satisfies certain properties, the transformation is good. The rest of the paper focuses on constructing such kernels for specific cases. These include Laplace, Uniform, and Erlang source distributions. Some analysis is for general classes of targets, but we only see complete analyses for univariate Gaussian and log-concave distributions (if I understand correctly).

The results appear non-trivial and, to the best of my knowledge, new. I am not an expert on this topic; I hope the other reviewers will weigh in on contribution here.

I felt there was gap between the results and the informal discussion. The abstract says "we provide computationally efficient procedures that approximately reduce uniform, Erlang, and Laplace location models to general target families." But the formal claims of computational efficiency depend on the target, and are only provided for specific targets (eg, Gaussians). Indeed, it seems that such a claim *cannot* be true, if you believe that there are families of distributions which are computationally hard to sample from. It is not clear to me that this approach even gives us an efficient transformation from multivariate Laplace to multivariate Gaussians; might the rejection sampling scale poorly with the dimension?

Does any prior work construct approximate, single-sample reductions for continuous $\Theta$? From the related work I gather the answer is "no," but it was not clear. If that is correct, it should be emphasized more.

Finally, I felt the presentation of the paper leaves much to be desired. I think the proofs would benefit from a handful of pictures to illustrate how they work for simple cases. The first part of the introduction does not address computation at all, and while the related work seems to discuss computational lower bounds it is not clear to me what the cited results imply about the question at hand.

In Section 3 in particular, I was a often confused by the presentation of the RK algorithm and the associated Lemma 1.
- We compute the Radon-Nikodym derivative $\frac{\mathrm{d}\bar{\mathcal{S}}}{\mathrm{d}\mathcal{P}}(y\mid x)$, but I do not see how to do this using only sampling access to $\mathcal{P}(\cdot\mid x)$.
- The algorithm uses a base measure $\mathcal{P}(\cdot\mid x)$, which is different from the source distribution $\mathcal{P}_\theta$. We also use $p(x)$, which is not a distribution. This notation caused some confusion for me.
- The algorithm takes some $y_0$ as initialization and Lemma 1 quantifies as "for all $y_0$...", but it looks like the algorithm doesn't use $y_0$ unless all proposals are accepted. Initializing a dummy $Y\sim \mathcal{P}(\cdot \mid x)$ would seem more natural to me, unless I misunderstand.
- Lemma 1 also says "for any $\epsilon\in (0,1)$ and $x\in \mathbb{X}$", but I do not see where these quantities show up. We do see "$x\mapsto \mathrm{RK}(x,N,M,y_0)$" but this refers to the algorithm, when we bound the running time.
- I had quite a bit of difficulty with the proof of Lemma 1. For example, the kernel $\widehat{T}(y\mid x)$ is defined in the main text in Eq (6), but in the appendix it appears without a pointer back to its definition. Also, for proving Eq. (49a), isn't it clear that the TV distance is upper bounded by the probability of failing to accept after $N$ steps? That's Eq. (100). I don't understand what purpose the other steps serve; a little more English text would go a long way here.

**Paper Award:**

No

---

### Author Rebuttal · Authors · 2024-11-24

# Response to e8Br [part 1]
Q. The formal claims of computational efficiency depend on the target, and are only provided for specific targets (eg, Gaussians). It seems that a reduction from uniform, Erlang, Laplace to a general target can't exist...

A. Thank you for pointing this out. We will revise the informal discussion to clarify that our concrete guarantees apply to key target distributions, including Gaussian and log-concave distributions. The reviewer is correct that there should not always exist an efficiently implementable Markov kernel that achieves small total variation (TV) deficiency for general target distributions, as some may violate the data processing inequality (see our responses to Reviewer DQo7) and others could contradict the planted clique conjecture (see Appendix F in the paper).

Having said that, we want to clarify that our reduction algorithm, denoted as $\mathsf{RK}$, and its guarantees (as stated in Lemma 1) are designed and proven for general target distributions. Furthermore, the signed kernel that achieves $\varsigma(\mathcal{U}, \mathcal{V}; \mathcal{S}^{*}) = 0$ is derived for general target distributions that only needs to satisfy mild regularity conditions (see Propositions 2, 4, and 9). The rationale behind presenting our results in this way is that the recipe for proving a general reduction is to first derive the corresponding signed kernel, and to then apply the algorithm $\mathsf{RK}$. The error guarantee can then be calculated using Eqs. (7) and (8). However, unless the obtained signed kernel is ''good'' in the sense of being close to a Markov kernel, the eventual deficiency bound may not be small. See our discussion of this point under Lemma 1 and in Appendix A.

Q. Does this reduction give us an efficient transformation from multivariate Laplace to multivariate Gaussians?

A. We can construct a reduction from the multivariate Laplace distribution in $\mathbb{R}^{d}$ with iid entries and covariance matrix $I_{d}$ to the multivariate Gaussian distribution with a covariance matrix of $\sigma^2 I_{d}$. This is accomplished by applying our reduction to each coordinate separately. By utilizing Theorem 3 and the triangle inequality, we see that this reduction achieves $\epsilon$-total variation (TV) deficiency when $\sigma$ is of the order $\log(d/\epsilon)$. Furthermore, the computational complexity for this process is $O(d \log(4d/\epsilon))$. However, a reduction for non-identity covariance remains an open question.

---

> ### Author Rebuttal · Authors · 2024-11-24
>
> # Response to e8Br [part 2]
> Q. Does any prior work construct approximate, single-sample reductions for continuous $\Theta$?
>
> A. Our work is the first that constructs approximate, single-sample reductions for continuous random variables and for continuous parameter set $\Theta$. We will highlight this further.
>
> Q. I think the proofs would benefit from a handful of pictures to illustrate how they work for simple cases. The first part of the introduction does not address computation at all...
>
> A. We will outline the main ideas of the proofs and include illustrations to show how they work. In fact, some proofs (see Sections E.2.1 and E.4.1) are already brief and straightforward, and we can move these to the main text. We will clarify the motivation for discussing computational lower bounds in the introduction. One important application of reductions is to establish computational hardness for high-dimensional statistical problems, which is why we reference these related works.
>
> Q. We compute the Radon-Nikodym derivative $\frac{\mathrm{d} \bar{\mathcal{S}}}{\mathrm{d} \mathcal{P}}(y|x)$, but I do not see how to do this using only sampling access to $\mathcal{P}(y|x)$.
>
> A. We will add the assumption that we can evaluate the Radon-Nikodym derivative $\frac{\mathrm{d} \bar{\mathcal{S}}}{\mathrm{d} \mathcal{P}}(y|x)$ efficiently, which is indeed the case for the concrete reductions in Section 4.
>
> Q. Base measure $\mathcal{P}(y|x)$, the source distribution $\mathcal{P}_{\theta}$, and $p(x)$ are confusing notation.
>
> A. We admit that this notation is confusing and we will make it clearer in the revision.
>
> Q. The algorithm takes some $y_{0}$ as initialization and Lemma 1 quantifies as "for all $y_{0}$ ...", but it looks like the algorithm doesn't use $y_{0}$ unless all proposals are accepted. Initializing a dummy would seem more natural to me, unless I misunderstand.
>
> A. The reviewer is correct that $y_0$ is the output of the algorithm $\mathsf{RK}$ when all the samples $Y_t$ are rejected. We set $y_0$ as an input to the algorithm so that users can choose a desired output in the event that all samples $Y_t$ are rejected. For instance, in the context of the privacy application (see Corollary 8 and Lemma 25), we choose $y_0$  to be the source sample $X_{\theta}$. This guarantees that the output $Y \leftarrow \mathsf{RK}(X_{\theta}, N, M, X_{\theta})$ has the same sub-exponential tail bound as the source sample $X_{\theta}$.

---

> > ### Comment · Reviewer_e8Br · 2024-11-26
> >
> > Thank you for your response, I found it helpful.

---

> > > ### Author Response · Authors · 2024-11-26
> > >
> > > Thank you for acknowledging our response, and for your questions and suggestions. We hope that our rebuttal has led you to view the paper more positively, but please let us know if there is anything else that we can clarify.

---

> ### Author Rebuttal · Authors · 2024-11-24
>
> # Response to e8Br [part 3]
>
> Q. Lemma 1 also says ''for any $\epsilon \in (0,1)$ and $x \in X$'', but I do not see where these quantities show up.
>
> A. We will remove the unnecessary part ''for any $\epsilon \in (0,1)$'' in the statement of Lemma 1.
>
> Q. I had quite a bit of difficulty with the proof of Lemma 1. For example, the kernel $\widehat{T}(y|x)$ is defined in the main text in Eq (6), but in the appendix it appears without a pointer back to its definition. Also, for proving Eq. (49a), isn't it clear that the TV distance is upper bounded by the probability of failing to accept after $N$ steps? That's Eq. (100). I don't understand what purpose the other steps serve; a little more English text would go a long way here.
>
> A. Thanks for pointing this out. We will add a pointer to the definition of $\widehat{T}(y|x)$ and provide additional explanations in the proofs to enhance their readability.
>
> In terms of the proof of Eq. (49a), the reviewer is correct that the main idea is to bound it by the probability of failing to accept after $N$ steps. Note that Eq. (100) is the probability of failing to accept for only one step and for a fixed $x$. To complete the proof of Eq. (49a), we need the probability of failing to accept after $N$ steps, which is equal to $g(x) = (1-p(x)/M)^{N}$ for a fixed $x$. The rest of the proof then averages this error $g(x)$ over the density of the source distribution $u(x;\theta)$.

---

### Author Rebuttal · Authors · 2024-11-25

# Response to DQo7 [part 1]

Q. Can the authors comment on whether a signed kernel with small error can be constructed for general source distributions?

A. Note that the optimal signed kernel can be defined as the solution of the infinite-dimensional convex optimization problem (Eq.(2b) in the paper)
\begin{align}
                    \inf_{\mathcal{S} \in \mathsf{S}(\mathbb{Y}| \mathbb{X}) }  \sup_{\theta \in \Theta}  \frac{1}{2} \Big\|\Big\| v(\cdot|\theta) - \int_{\mathbb{X}} \mathcal{S}(\cdot|x) \cdot u(x|\theta) \mathrm{d}x \Big\|\Big\|_{1}.
\end{align}
It exists in a wide range of scenarios and the objective value of the above problem should be small (or zero) in these cases. However, it is not guaranteed that this optimal signed kernel is close to a Markov kernel; in other words, finding the closest Markov kernel induces large error.  We provide a concrete example below.

Let the source be given by the Bernoulli location model $\mathsf{Ber}(\theta)$, the target be given by the Gaussian location model $\mathcal{N}(\theta,\sigma^2)$, and the parameter set $\Theta = (\theta_1, \theta_2)$ for two fixed $\theta_1,\theta_2 \in [0,1]$. Note that $\mathbb{X} = (0,1)$ and $v(y|\theta) = \exp\{-(y-\theta)^2/(2\sigma^2)\}/(\sqrt{2\pi}\sigma)$, then the solution
\begin{align} \mathcal{S}(y|0) = \frac{\theta_2 \cdot v(y|\theta_1) - \theta_1\cdot v(y|\theta_2)}{\theta_2 - \theta_1}, \quad \mathcal{S}(y|1) = \frac{(1-\theta_2) \cdot v(y|\theta_1) - (1-\theta_1)\cdot v(y|\theta_2)}{\theta_1 - \theta_2}
\end{align}
attains zero objective value of the above optimization problem. However, it is impossible to have a good Markov kernel when $\sigma$ is small. In fact, any Markov kernel that attains small TV-deficiency violates the data processing inequality. Suppose for the sake of contradiction we have a reduction $\mathsf{RK}$ (a Markov kernel) that can transfer $\mathsf{Ber}(\theta)$ to $\mathcal{N}(\theta)$ with $\epsilon$-TV deficiency. Let $X_{\theta_i} \sim \mathsf{Ber}(\theta_i)$ and $Y_{\theta_i} = \mathsf{RK}(X_{\theta_i})$ for $i=1,2$.
Then we have $|| X_{\theta_1} - X_{\theta_2}|| = |\theta_1 - \theta_2|$, where $|| \cdot ||$ denotes the TV distance. And
\begin{align} || Y_{\theta_1} - Y_{\theta_2}|| \geq || \mathcal{N}(\theta_1,\sigma^2) - \mathcal{N}(\theta_2,\sigma^2)  || - ||\mathcal{N}(\theta_1,\sigma^2) - Y_{\theta_1}  || - ||\mathcal{N}(\theta_2,\sigma^2) - Y_{\theta_2}  ||
\geq \min(1/200, |\theta_1 - \theta_2|/(5\sigma) )- 2\epsilon.
\end{align}

---

> ### Author Rebuttal · Authors · 2024-11-25
>
> # Response to DQo7 [part 2]
>
> A. In the above display, we use the triangle inequality in the first step, and in the step step we apply Theorem 1.3 of paper "The total variation distance between high-dimensional Gaussians with the same mean" by Devroye et al. Consequently, when $\sigma = 1/10$, $2\epsilon < |\theta_1 - \theta_2| \leq 1/400$, we obtain$ || Y_{\theta_1} - Y_{\theta_2} || > || X_{\theta_1} - X_{\theta_2} ||$. This violates the data processing inequality for total variation.
>
> Q. Could the authors comment on how good the reductions are? Is there a lower bound on the number of oracle calls to the pointwise evaluation on the target model?
>
> A. In our concrete reductions, to achieve $\epsilon$-TV deficiency, the algorithm $\mathsf{RK}$ only needs $O(\log(4/\epsilon))$ pointwise evaluations of $\frac{\mathrm{d} \bar{\mathcal{S}}}{\mathrm{d} \mathcal{P}}(y|x)$ and $O(\log(4/\epsilon))$ samples from base measure $\mathcal{P}(\cdot|x)$. Note that the Radon–Nikodym derivatives (see, e.g., Eq.(57), Eq.(82)) have explicit and clean formulas, so pointwise evaluation is cheap. Moreover, we choose the base measure $\mathcal{P}(\cdot|x)$ to be either a Gaussian or log-concave distribution (see Theorems 3, 4, 5), so sampling is also efficient. It is not clear to us whether there is a lower bound on the complexity, our whether our complexity bound of  $O(\log(4/\epsilon))$ can be improved further.
>
> Besides the theoretical guarantees, the reductions have appealing empirical performance and are readily implementable. Please see Appendix B for the experimental results.
>
> Q. While the paper list a few applications, the reduction approach doesn't seem to lead to new results in the described applications.
>
> A. Please note that all the results we developed in our applications are novel contributions to the literature. Specifically, our reduction is the first to successfully transfer a sample from the symmetric mixture of linear experts model, where the additive noise is uniform, to a sample from the phase retrieval model with Gaussian additive noise in the finite signal-to-noise ratio regime. Additionally, our reduction is also the first to convert a Laplace mechanism into a Gaussian mechanism without the need to de-privatize the query. Finally, as demonstrated in Proposition 15, our reduction enables us to provide sharp error guarantees for signal denoising problems when the additive noise is non-Gaussian, which is another new finding in the literature.

---

> ### Author Rebuttal · Authors · 2024-11-25
>
> # Response to DQo7 [part 3]
>
> Q. Corollary 8: It seems the accuracy here is even worse than the sum of the variance of the original Laplace distribution and the targeted Gaussian distribution. So the usefulness of the result is less clear.
>
> A. Recall that we have $g(\boldsymbol{X})\sim \mathsf{Lap}(f(\boldsymbol{X}),b)$ and our goal is to transform $g(\boldsymbol{X})$ into a random variable that is $\delta$-close in total variation to one drawn from the distribution $\mathcal{N}(f(\boldsymbol{X}), \sigma^2)$. Our reduction can achieve this goal for $\sigma^{2} = 2b^2 \log(12/\delta)$. The key takeaway is that the variance blow-up is only $\textbf{logarithmic}$ in $1/\delta$. As stated in Corollary 8, the resulting mechanism has accuracy
> $$
> \sqrt{2b^2\log(12/\delta) + 2b^2 + \frac{b^2}{4} \cdot \delta \log^{3/2}(12/\delta) }.
> $$
> We agree with the reviewer that this slightly exceeds the sum of the variance of the original Laplace distribution and the targeted Gaussian distribution by an additive term $\frac{b^2}{4} \cdot \delta \log^{3/2}(12/\delta)$. However, since $\delta$ is set to be very small (close to 0), the term $\frac{b^2}{4} \cdot \delta \log^{3/2}(12/\delta)$ is ignorable compared to the other two terms in the above display.
>
> To illustrate the usefulness of the result, compare our reduction with the natural plugin approach  $\mathsf{K}_{\mathsf{plugin}}(g(\boldsymbol{X})) =  g(\boldsymbol{X}) + \sigma Z$, where $Z \sim \mathcal{N}(0,1)$. By construction, for $\sigma$ large enough, this should take a Laplace random variable to something resembling a Gaussian random variable of the same mean.
>
> But by Proposition 16 of the paper, we obtain
> $$
> || \mathsf{K}_{\mathsf{plugin} }( g(\boldsymbol{X}) ) - \mathcal{N}(f(\boldsymbol{X}), \sigma^2 ) || \gtrsim \frac{b^{2}/\sigma^2}{1+b^2/\sigma^2}.
> $$
>
> Consequently, to guarantee $\delta$-TV deficiency, we must set $\sigma \geq b/\delta$ in the plugin approach, which increases $\textbf{polynomially}$ with $1/\delta$. The accuracy of $\mathsf{K}_{\mathsf{plugin}}(g(\boldsymbol{X}))$ is $\sqrt{2b^2+b^2/\delta^2}$, which is much worse than our guarantee.

---

### Author Rebuttal · Authors · 2024-11-25

# Response to DPEr
We appreciate the reviewer's positive feedback on our paper. Answers to specific questions are below:

Q. Cite "Statistical-Computational Tradeoffs in Mixed Sparse Linear Regression" by Arpino, Venkataramanan (COLT 2023)

A. Thanks for bringing this related work to our attention, and we will definitely cite this paper in the revision.

Q. How do your results relate to the paper: "Hypothesis testing with low-degree polynomials in the Morris class of exponential families" by Dmitriy Kunisky?

A. Thanks for bringing this related work to our attention, and we will definitely cite this paper in the revision. This related work establishes a partial ordering of distributions in terms of computational hardness via the low-degree method. Indeed, they show
	$$
		\mathsf{Bernoulli} \geq  \mathsf{Binomial}  \geq  \mathsf{Gaussian} \geq \mathsf{Poisson} \geq \mathsf{Exponential},
	$$
	where ''$\geq$'' denotes greater computational difficulty. Our reduction complements this ordering in two key ways. First, our reduction algorithm runs in polynomial time in terms of the dimension of the problem and the TV deficiency. This allows us to establish computational hardness results for the class of all polynomial-time methods, thereby extending beyond the low-degree method. For this class, our concrete reductions from Uniform, Laplace, Erlang to Gaussian establish the following partial ordering
	$$
		\mathsf{Uniform} \geq \mathsf{Gaussian}, \quad \mathsf{Laplace} \geq \mathsf{Gaussian}, \quad \mathsf{Erlang} \geq \mathsf{Gaussian}.
	$$

---

> ### Comment · Reviewer_DPEr · 2024-11-28
> **Response**
>
> I understand, thank you for addressing my questions.

---

> > ### Author Response · Authors · 2024-11-28
> >
> > Thank you for acknowledging our response. Please let us know if there is anything else that we can clarify.

---

### Meta-Review · Area_Chair_NydD · 2024-12-08

**Recommendation:** Accept
**Confidence:** 5

**Metareview:**

All three referees gave this paper positive recommendations. The authors have responded to initial reviews thoughtfully, and the reviewers are satisfied with the rebuttal.

**Paper Award:**

No